# Paclitaxel-Associated Mechanical Sensitivity and Neuroinflammation Are Sex-, Time-, and Site-Specific and Prevented through Cannabigerol Administration in C57Bl/6 Mice

**DOI:** 10.3390/ijms25084277

**Published:** 2024-04-12

**Authors:** Hongbo Li, Sara Jane Ward

**Affiliations:** Center for Substance Abuse Research, Department of Neural Sciences, Lewis Katz School of Medicine, Temple University, Philadelphia, PA 19140, USA; hongboli@temple.edu

**Keywords:** pain, cannabinoid, cannabigerol, sex differences, PAG, neuroinflammation, paclitaxel

## Abstract

Chemotherapy-induced peripheral neuropathy (CIPN) is one of the most prevalent and dose-limiting complications in chemotherapy patients. One identified mechanism underlying CIPN is neuroinflammation. Most of this research has been conducted in only male or female rodent models, making direct comparisons regarding the role of sex differences in the neuroimmune underpinnings of CIPN limited. Moreover, most measurements have focused on the dorsal root ganglia (DRG) and/or spinal cord, while relatively few studies have been aimed at characterizing neuroinflammation in the brain, for example the periaqueductal grey (PAG). The overall goals of the present study were to determine (1) paclitaxel-associated changes in markers of inflammation in the PAG and DRG in male and female C57Bl6 mice and (2) determine the effect of prophylactic administration of an anti-inflammatory cannabinoid, cannabigerol (CBG). In Experiment 1, male and female mice were treated with paclitaxel (8–32 mg/kg/injection, Days 1, 3, 5, and 7) and mechanical sensitivity was measured using Von Frey filaments on Day 7 (Cohort 1) and Day 14 (Cohort 2). Cohorts were euthanized on Day 8 or 15, respectively, and DRG and PAG were harvested for qPCR analysis of the gene expression of markers of pain and inflammation *Aig1*, *Gfap*, *Ccl2*, *Cxcl9*, *Tlr4*, *Il6*, and *Calca*. In Experiment 2, male and female mice were treated with vehicle or 10 mg/kg CBG i.p. 30 min prior to each paclitaxel injection. Mechanical sensitivity was measured on Day 14. Mice were euthanized on Day 15, and PAG were harvested for qPCR analysis of the gene expression of *Aig1*, *Gfap*, *Ccl2*, *Cxcl9*, *Tlr4*, *Il6*, and *Calca*. Paclitaxel produced a transient increase in potency to produce mechanical sensitivity in male versus female mice. Regarding neuroinflammation, more gene expression changes were apparent earlier in the DRG and at a later time point in the PAG. Also, more changes were observed in females in the PAG than males. Overall, sex differences were observed for most markers at both time points and regions. Importantly, in both the DRG and PAG, most increases in markers of neuroinflammation and pain occurred at paclitaxel doses higher than those associated with significant changes in the mechanical threshold. Two analytes that demonstrated the most compelling sexual dimorphism and that changed more in males were *Cxcl9* and *Ccl2*, and *Tlr4* in females. Lastly, prophylactic administration of CBG protected the male and female mice from increased mechanical sensitivity and female mice from neuroinflammation in the PAG. Future studies are warranted to explore how these sex differences may shed light on the mechanisms of CIPN and how non-psychoactive cannabinoids such as CBG may engage these targets to prevent or attenuate the effects of paclitaxel and other chemotherapeutic agents on the nervous system.

## 1. Introduction

Paclitaxel has been used clinically to treat various cancers for the past 30 years. Unfortunately, paclitaxel treatment can also be associated with pain, including both sub-acute pain syndrome and long-term chemotherapy-induced peripheral neuropathy (CIPN). CIPN is one of the most prevalent and dose-limiting complications in chemotherapy patients. It is characterized by pain and numbness in a stocking and glove distribution. CIPN affects a significant percentage of cancer survivors, with symptoms lasting for weeks or months, or even years after treatment has been completed. Importantly, women experience clinical as well as experimental pain differently than men, including increased sensitivity to pain and an increase in the development of chronic pain [1]. It is also the case that women experience more severe adverse toxic effects of chemotherapy treatment compared with men, although we are unaware of broad clinical analyses regarding sex differences for CIPN [2].

The mechanisms underlying CIPN are not fully understood; however, the development of rodent models of pain associated with paclitaxel exposure have greatly advanced our understanding since their development in rats in the early 2000s [3,4,5]. With the concomitant advance in genetic animal models, use of mouse models of CIPN in addition to rat models has also significantly increased. Using these models, growing evidence indicates that one mechanism by which paclitaxel exposure leads to chronic neuropathic pain is its effects on peripheral neuroinflammation. Several studies have shown that paclitaxel exposure drives dorsal root ganglia (DRG) neuroinflammation by resident and recruited immune cells (for review see [6]). For example, paclitaxel increases the expression of pro-inflammatory factors like *Tlr4* [7], TNF-α [8], IL-1β [9], IL-6 [10], and *Ccl2/Mcp1* [11] in rodent models of paclitaxel-induced peripheral neuropathy. Taken together, these and other peripheral neuroinflammatory events may contribute significantly to central sensitization and the development and/or maintenance of neuropathic pain. It should be noted that the majority of these data come from preclinical models, and there is limited evidence available from human studies as to what extent neuroinflammatory responses contribute to neuropathic pain associated with paclitaxel treatment in patients.

Most of this rodent work has been conducted in only male or female models, making direct comparisons regarding the role of sex differences in neuroimmune underpinnings of paclitaxel-associated neuropathic pain limited (but see [10,12,13,14]). Considering the recent interest in sex differences in chronic pain development and immune cell activation, it is proposed that neuroimmune mechanisms vary between sexes. Moreover, most measurements of paclitaxel-associated neuroinflammation have focused on the DRG and/or spinal cord, while relatively few studies have aimed at characterizing neuroinflammation in the brain following paclitaxel administration (anterior cingulate cortex [15]; prefrontal cortex [16]; S1 cortex [17]; periaqueductal grey (PAG) and thalamus [18]). This is important, as the CNS generally, and the brain particularly, plays a previously under-recognized role in the pathophysiology of human CIPN [19]. This perspective is consistent with the fact that the pharmacological intervention currently associated with the most positive impact on CIPN is duloxetine, which acts in the brain as a serotonin-norepinephrine reuptake inhibitor [20]. Other medications, such as gabapentin, tricyclic antidepressants, creams, and certain natural products, are also used but share poor efficacy and/or intolerable adverse effects [21].

There is growing patient and research interest in cannabinoid constituents for the treatment of CIPN (see [22] for review). For example, we and others have demonstrated that both Δ^9^-tetrahydrocannabinol (THC) and cannabidiol (CBD), alone and in combination, can prevent the development of mechanical sensitivity associated with paclitaxel administration in male [23,24] or female [25,26] C57Bl/6 mice. Most recently, it was demonstrated that another phytocannabinoid, cannabigerol (CBG), significantly reduced mechanical hypersensitivity in a mouse model of cisplatin-associated peripheral neuropathy [27,28].

Therefore, the rationale for this study was to more comprehensively characterize the development of behavioral and neuroinflammatory consequences of paclitaxel exposure in both male and female mice and how they are impacted by the prophylactic administration of CBG. In Experiment 1, male and female mice were treated with a range of paclitaxel doses to determine potency differences in the ability of paclitaxel to produce mechanical sensitivity and neuroinflammation in the DRG or PAG over time. In Experiment 2, male and female mice were treated with vehicle or 10 mg/kg CBG to determine whether this prophylactic treatment strategy would prevent the development of mechanical sensitivity and neuroinflammation in the PAG.

## 2. Results

### 2.1. Experiment 1. Paclitaxel-Associated Mechanical Sensitivity in Male and Female C57Bl/6 Mice

Mechanical sensitivity was measured in male and female C57Bl/6 mice on Day 7 and Day 14 in two separate cohorts of mice (Figure 1). In the Day 7 cohort, paclitaxel produced dose-dependent increases in mechanical sensitivity in male and female mice. In the males, one-way ANOVA revealed a significant effect of paclitaxel [F_(4,35)_ = 8.105], *p* < 0.0009, with a Dunnett’s multiple comparison test showing that the 16, 24, and 32 mg/kg doses were statistically significant from vehicle. In the females, one-way ANOVA revealed a significant effect of paclitaxel [F_(4,35)_ = 5.454], *p* < 0.0016, with Dunnett’s multiple comparison test showing that the 24, and 32 mg/kg doses were statistically significant from vehicle. In the Day 14 cohort, paclitaxel produced dose-dependent increases in mechanical sensitivity in male and female mice. In the males, one-way ANOVA revealed a significant effect of paclitaxel [F_(4,33)_ = 5.614], *p* < 0.0015, with Dunnett’s multiple comparison test showing that the 8.0, 16, 24, and 32 mg/kg doses were statistically significant from vehicle. Additionally, two of the male mice in the 32 mg/kg treatment group were found dead in their home cages on Day 8. In the females, one-way ANOVA revealed a significant effect of paclitaxel [F_(4,36)_ = 13.33], *p* < 0.0001, with a Dunnett’s multiple comparison test showing that the 16, 24, and 32 mg/kg doses were statistically significant from vehicle. Regarding sex differences, two-way ANOVA on Day 7 data showed a significant effect of sex [F_(1,70)_ = 5.524, *p* = 0.0216] and of dose [F_(4,70)_ = 10.05, *p* < 0.0001], but no interaction [F_(4,70)_ = 1.474, *p* = 0.2194]. Šidák’s multiple comparison test showed that the 16 mg/kg dose of paclitaxel showed a significantly larger effect on mechanical sensitivity in male than in female mice. Two-way ANOVA on Day 14 data showed no significant effect of sex [F_(1,68)_ < 1.0] but a significant effect of dose [F_(4,68)_ = 13.02, *p* < 0.0001], and no interaction [F_(4,68)_ = 2.456, *p* = 0.0531]. Therefore, while paclitaxel was initially more potent in males regarding the production of mechanical sensitivity, by Day 14, paclitaxel was equipotent and effective at this endpoint in males and females.

### 2.2. Paclitaxel Increases Expression of Aif1 in DRG and PAG of Male and Female C57Bl/6 Mice

The *Aif1* gene codes for the Iba1 protein. The Iba1 protein is specifically upregulated in macrophages and microglia upon activation, for example following injury to the nervous system. Following paclitaxel administration, *Aif1* is generally dose-dependently elevated in the DRG and PAG of male and female mice (Figure 2A–H). One-way ANOVAs are as follows: Males Day 8 DRG [F_(4,16)_ = 3.280, *p* < 0.0382], Dunnett’s shows 32 mg/kg statistically significant from vehicle; Males Day 15 DRG [F_(4,16)_ = 4.342, *p* < 0.0116], Dunnett’s shows 8.0 and 32 mg/kg statistically significant from vehicle; Females Day 8 DRG [F_(4,16)_ = 4.428, *p* < 0.0199], Dunnett’s shows no posttest significance; Females Day 15 DRG [F_(4,16)_ = 18.37, *p* < 0.0006], Dunnett’s shows 24 and 32 mg/kg statistically significant from vehicle; Males Day 8 PAG [F_(4,13)_ < 1.0]; Males Day 15 PAG [F_(4,20)_ = 5.976, *p* < 0.0025], Dunnett’s shows 8.0, 16, and 24 mg/kg statistically significant from vehicle; Females Day 8 PAG [F_(4,19)_ = 4.817, *p* < 0.0075], Dunnett’s shows 16 and 32 mg/kg statistically significant from vehicle; Females Day 15 PAG [F_(4,19)_ = 6.663, *p* < 0.0016], Dunnett’s shows 32 mg/kg statistically significant from vehicle. As seen in Table 1, therefore, Iba1 activation is elevated in males and females in both the DRG and PAG. These effects were in some cases observed at the lowest dose that produced mechanical sensitivity, but in other cases required higher doses than those required to produce allodynia. Further statistical analysis with two-way ANOVA demonstrated a significant effect of sex in the DRG on Day 8, but not Day 15, or in the PAG (see Table 2 for F and *p* values).

### 2.3. Paclitaxel Increases Expression of Gfap in DRG but Not PAG of Male and Female C57Bl/6 Mice

The *Gfap* gene codes for the GFAP protein, a marker for astrocytes that is upregulated with injury and inflammatory states. Following paclitaxel administration, *Gfap* expression is dose dependently elevated in the DRG but not the PAG of male and female mice (Figure 3A–H). One-way ANOVAs are as follows: Males Day 8 DRG [F_(4,17)_ = 7.212, *p* < 0.0014], Dunnett’s shows 32 mg/kg statistically significant from vehicle; Males Day 15 DRG [F_(4,18)_ = 2.358, *p* < 0.0923]; Females Day 8 DRG [F_(4,15)_ = 5.163, *p* < 0.0081], Dunnett’s shows 32 mg/kg statistically significant from vehicle; Females Day 15 DRG [F_(4,12)_ = 5.671, *p* < 0.0084], Dunnett’s shows 32 mg/kg statistically significant from vehicle; Males Day 8 PAG [F_(4,13)_ = 2.101, *p* = 0.1389]; Males Day 15 PAG [F_(4,20)_ < 1.0]; Females Day 8 PAG [F_(4,19)_ = 2.450, *p* = 0.0814]; Females Day 15 PAG [F_(4,18)_ = 1.121, *p* = 0.3776]. As seen in Table 1, therefore, GFAP activation is elevated in males and females in the DRG but not the PAG, and these effects required higher doses than those required to produce allodynia. A further statistical analysis with two-way ANOVA demonstrated a significant effect of sex in the DRG on Days 8 and 15 and in the PAG on Day 8 (see Table 2 for F and *p* values).

### 2.4. Paclitaxel Increases Expression of Ccl2 in DRG and PAG of Male and Female C57Bl/6 Mice

The *Ccl2* gene codes for the CCL2 protein, a small cytokine that recruits monocytes, memory T cells, and dendritic cells to the sites of inflammation. It is also known as monocyte chemoattractant protein-1 (MCP-1). Following paclitaxel administration, *CCL2* expression is elevated in the DRG and PAG of female and, to a lesser extent, male mice (Figure 4A–H). One-way ANOVAs are as follows: Males Day 8 DRG [F_(4,17)_ = 3.073, *p* < 0.0448], Dunnett’s shows 32 mg/kg statistically significant from vehicle; Males Day 15 DRG [F_(4,18)_ = 1.993, *p* = 0.1388]; Females Day 8 DRG [F_(4,16)_ = 41.78, *p* < 0.0001], Dunnett’s shows significance at the 16 mg/kg dose; Females Day 15 DRG [F_(4,17)_ = 6.711, *p* < 0.0020], Dunnett’s shows 32 mg/kg statistically significant from vehicle; Males Day 8 PAG [F_(4,20)_ = 2.207, *p* = 0.1050]; Males Day 15 PAG [F_(4,20)_ = 2.172, *p* < 0.1092]; Females Day 8 PAG [F_(4,17)_ = 2.921, *p* = 0.0423], Dunnett’s shows 32 mg/kg statistically significant from vehicle; Females Day 15 PAG [F_(4,20)_ = 3.230, *p* < 0.0337], Dunnett’s shows 24 mg/kg statistically significant from vehicle. As seen in Table 1, therefore, *Ccl2* expression is elevated in males and females in the DRG, but only females in the PAG; only the effect in females at Day 8 in the DRG is at the minimal effective dose to produce mechanical sensitivity. Further statistical analysis with two-way ANOVA demonstrated a significant effect of sex in the DRG on Days 8 and 15 and in the PAG on Day 15 (see Table 2 for F and *p* values).

### 2.5. Paclitaxel Increases Expression of Cxcl9 in DRG but Not PAG in Male but Not Female C57Bl/6 Mice

The *Cxcl9* gene codes for the CXCL9 protein, a small cytokine that plays a role in immune cell migration, differentiation, and activation of cytotoxic lymphocytes (CTLs), natural killer (NK) cells, NKT cells, and macrophages. Following paclitaxel administration, *Cxcl9* expression is only statistically significantly elevated in the DRG of male mice and was out of the detectable range in the PAG of male, but not female mice (Figure 5A–F). One-way ANOVAs are as follows: Males Day 8 DRG [F_(4,13)_ = 5.434, *p* < 0.0085], Dunnett’s shows 16 mg/kg statistically significant from vehicle; Males Day 15 DRG [F_(4,14)_ = 4.764, *p* < 0.0123], Dunnett’s shows 24 and 32 mg/kg statistically significant from vehicle; Females Day 8 DRG [F_(4,16)_ = 1.755, *p* = 0.1873]; Females Day 15 DRG [F_(4,17)_ = 1.702, *p* = 0.1959]; Females Day 8 PAG [F_(4,13)_ = 1.760, *p* = 0.1971]; Females Day 15 PAG [F_(4,14)_ = 2.752, *p* < 0.0704]. Table 1 summarizes that *Cxcl9* expression is selectively increased in males in the DRG, including at the minimum effective dose to produce mechanical sensitivity. A further statistical analysis with two-way ANOVA demonstrated a significant effect of sex in the DRG on Days 8 and 15 (see Table 2 for F and *p* values).

### 2.6. Paclitaxel Increases Expression of Tlr4 in DRG of Male Mice and DRG and PAG of Female C57Bl/6 Mice

The *Tlr4* gene codes for the TLR4 protein, a transmembrane protein member of the toll-like receptor family. Its activation leads to NF-κB signaling and inflammatory cytokine production in myeloid cells and is also expressed on neurons. Following paclitaxel administration, *Tlr4* expression is elevated in the DRG of male and female mice and the PAG of female mice (Figure 6A–H). One-way ANOVAs are as follows: Males Day 8 DRG [F_(4,17)_ = 6.552, *p* = 0.0022], Dunnett’s shows 16 and 32 mg/kg statistically significant from vehicle; Males Day 15 DRG [F_(4,19)_ < 1.0]; Females Day 8 DRG [F_(4,19)_ = 12.30, *p* < 0.0001], Dunnett’s shows significance at the 16 mg/kg dose; Females Day 15 DRG [F_(4,17)_ = 2.976, *p* = 0.0515]; Males Day 8 PAG [F_(4,14)_ < 1.0, *p* = 0.8566]; Males Day 15 PAG [F_(4,20)_ = 2.340, *p* = 0.0901]; Females Day 8 PAG [F_(4,19)_ = 4.543, *p* = 0.0096], Dunnett’s shows 16 mg/kg statistically significant from vehicle; Females Day 15 PAG [F_(4,18)_ = 4.517, *p* = 0.0106], Dunnett’s shows 16 mg/kg statistically significant from vehicle. As seen in Table 1, *Tlr4* expression is elevated in males and females in the DRG, but only females in the PAG; only the effect in females is at the minimal effective dose to produce mechanical sensitivity. A further statistical analysis with two-way ANOVA demonstrated a significant effect of sex in the DRG on Day 15 and in the PAG on Day 8 (see Table 2 for F and *p* values).

### 2.7. Paclitaxel Increases Expression of Il6 in DRG of Male Mice and DRG and PAG of Female C57Bl/6 Mice

The *Il6* gene codes for the IL-6 protein, an interleukin that is released by macrophages and acts as a pro-inflammatory cytokine. Following paclitaxel administration, *Il6* expression is elevated in the DRG of male and female mice and the PAG of female mice (Figure 7A–H). One-way ANOVAs are as follows: Males Day 8 DRG [F_(4,13)_ = 9.811, *p* = 0.0007], Dunnett’s shows 32 mg/kg statistically significant from vehicle; Males Day 15 DRG [F_(4,18)_ < 1.0]; Females Day 8 DRG [F_(4,14)_ = 8.605, *p* = 0.0010], Dunnett’s shows significance at the 16 mg/kg dose; Females Day 15 DRG [F_(4,14)_ = 4.378, *p* = 0.0167], Dunnett’s shows significance at the 24 mg/kg dose; Males Day 8 PAG [F_(4,18)_ < 1.0]; Males Day 15 PAG [F_(4,15)_ < 1.0]; Females Day 8 PAG [F_(4,19)_ < 1.0]; Females Day 15 PAG [F_(4,16)_ = 2.347, *p* = 0.0985]. Table 1 summarizes that *Il6* expression is selectively increased in the DRG, including in females at Day 8 at the minimum effective dose to produce mechanical sensitivity. Further statistical analysis with two-way ANOVA demonstrated a significant effect of sex in the DRG and PAG on Day 15 (see Table 2 for F and *p* values).

### 2.8. Paclitaxel Increases Expression of Calca in DRG in Male Mice and PAG in Male and Female C57Bl/6 Mice

The *Calca* gene codes for the CGRP protein; CGRP is produced in both peripheral and central neurons and functions in the transmission of nociception. Following paclitaxel administration, *Calca* expression is elevated in the DRG but not the PAG of the male mice, and both the DRG and PAG of the female mice (Figure 8A–H). One-way ANOVAs are as follows: Males Day 8 DRG [F_(4,17)_ = 3.058, *p* = 0.0455], Dunnett’s shows 16 and 32 mg/kg statistically significant from vehicle; Males Day 15 DRG [F_(4,19)_ = 1.044, *p* = 0.4107]; Females Day 8 DRG [F_(4,20)_ = 6.148, *p* = 0.0021], Dunnett’s shows 16 mg/kg statistically significant from vehicle; Females Day 15 DRG [F_(4,19)_ = 4.592, *p* = 0.0092], Dunnett’s shows 16 mg/kg statistically significant from vehicle; Males Day 8 PAG [F_(4,14)_ = 1.087, *p* = 0.4004]; Males Day 15 PAG [F_(4,20)_ = 1.439, *p* = 0.2580]; Females Day 8 PAG [F_(4,17)_ = 1.987, *p* = 0.1423]; Females Day 15 PAG [F_(4,19)_ = 6.663, *p* = 0.0016], Dunnett’s shows 32 mg/kg statistically significant from vehicle. Table 1 summarizes that CGRP expression is selectively increased in males and females at Day 8 in the DRG, at minimum effective doses that produce mechanical sensitivity. A further statistical analysis with two-way ANOVA demonstrated a significant effect of sex in the DRG and PAG on Day 15 (see Table 2 for F and *p* values).

### 2.9. Prophylactic CBG Administration Prevents Mechanical Sensitivity Associated with Paclitaxel Exposure in Male and Female C57Bl/6 Mice

Mechanical sensitivity was measured in male and female C57Bl/6 mice at baseline and on Day 14. Mice were treated with vehicle or CBG 30 min prior to vehicle or paclitaxel on Days 1, 3, 5, and 7. Different (8.0, 24 mg/kg) and overlapping (16 mg/kg) doses of paclitaxel were tested in males and females based on the effects observed in Experiment 1 where paclitaxel was more potent in male versus female mice. The results demonstrate that the prophylactic administration of CBG protected the female and male mice from increased mechanical sensitivity (Figure 9). In the male mice, two-way ANOVA showed a significant effects of paclitaxel [F_(2,34)_ = 6.430, *p* = 0.0037], CBG [F_(1,34)_ = 10.56, *p* = 0.0023], but there was no interaction. Sidak’s multiple comparison test showed a significant effect of CBG on 16 mg/kg paclitaxel. In the female mice, two-way ANOVA showed a significant effect of paclitaxel [F_(2,42)_ = 14.14, *p* = 0.0001], a significant effect of CBG [F_(1,42)_ = 6.275, *p* = 0.0130], but no significant interaction. Sidak’s multiple comparison test showed a significant effect of CBG on 24 mg/kg paclitaxel.

### 2.10. Prophylactic CBG Administration nor Paclitaxel Influenced Expression of Inflammation and Pain Markers in the PAG of Male C57Bl/6 Mice

Male mice were treated with vehicle or CBG (10 mg/kg i.p.); 30 min prior to vehicle or paclitaxel (8 or 16 mg/kg i.p.) on Days 1, 3, 5, and 7, and on Day 15, they were euthanized, and PAGs were harvested for RT-PCR analysis. The results demonstrated that, as in the previous experiment with paclitaxel alone, neither paclitaxel alone nor in combination with CBG administration affected the expression of any of the genes measured (Figure 10A–F). The results for individual one-way ANOVAs are as follows: *Aif1* [F_(4,19)_ < 1.0]; *Ccl2* [F_(4,17)_ = 2.064]; *Calca* [F_(4,18)_ = 2.693, *p* = 0.0641]; *Tlr4* [F_(4,19)_ = 1.081]; *Gfap* [F_(4,20)_ < 1.0]; *Il6* [F_(4,19)_ = 1.014].

### 2.11. Prophylactic CBG Administration Prevents Increased Expression of Inflammation and Pain Markers in the PAG of Female C57Bl/6 Mice

Female mice were treated with vehicle or CBG (10 mg/kg i.p.) 30 min prior to vehicle or paclitaxel (16 or 24 mg/kg i.p.) on Days 1, 3, 5, and 7, and on Day 15 they were euthanized and PAGs were harvested for RT-PCR analysis. The 16 mg/kg dose of paclitaxel again significantly increased expression of *Tlr4*, while the 24 mg/kg dose of paclitaxel increased expression of *Aif1*, *Ccl2*, and *Calca*. Results demonstrated that CBG administration prevented increased expression of *Aif1*, *Ccl2*, *Calca*, and *Tlr4* associated with paclitaxel exposure (Figure 11A–G). Results for individual one-way ANOVAs are as follows: *Aif1* [F_(4,19)_ = 6.691, *p* = 0.0015], with Dunnett’s multiple comparison test showing 24 mg/kg paclitaxel statistically significant from vehicle; *Ccl2* [F_(4,16)_ = 3.530, *p* = 0.0301], with Dunnett’s multiple comparison test showing 24 mg/kg paclitaxel statistically significant from vehicle; *Calca* [F_(4,19)_ = 3.308, *p* = 0.0323], with Dunnett’s multiple comparison test showing 24 mg/kg paclitaxel statistically significant from vehicle; *Tlr4* [F_(4,19)_ = 8.602, *p* = 0.0004], with Dunnett’s multiple comparison test showing 16 mg/kg paclitaxel statistically significant from vehicle; *Gfap* [F_(4,18)_ = 1.638, *p* = 0.2082]; *Il6* [F_(4,14)_ = 1.928, *p* = 0.1616]; *Cxcl9* [F_(4,19)_ = 1.450, *p* = 0.2565].

## 3. Discussion

To test our hypothesis that the prophylactic administration of CBG would attenuate the development of mechanical sensitivity and associated neuroinflammation in male and female C57Bl/6 mice, we first sought determine whether we observed any sex differences regarding the effects of paclitaxel exposure itself on these endpoints. Indeed, paclitaxel produced a quicker onset of behavioral toxicity in males compared with females, but there were equal observable effects over time. To determine whether these changes and sex differences in mechanical sensitivity in male and female mice were associated with neuroinflammation in the DRG or PAG, we measured the expression of mRNA for *Aif-1*, *Gfap*, *Ccl-2*, *Cxcl9*, *Tlr-4*, *Il6*, and *Calca* (Table 1). In general, more neuroinflammatory changes were observed in the DRG than in the PAG. In the DRG, more changes were apparent on Day 8 than Day 15, and sex differences were observed for the expression of all genes tested on Days 8 and/or 15. In the PAG, in contrast to the DRG, more changes were apparent at Day 15 than Day 8, and more changes were observed in females as compared to males. Again, significant sex differences were observed. Importantly, in both the DRG and PAG, most increases in markers of neuroinflammation and pain occurred at paclitaxel doses higher than those associated with significant changes in the mechanical threshold. Overall, the most sensitive neuroinflammatory measurements that were closely associated by dose to mechanical sensitivity changes were those observed on Day 8 in the DRG of female mice. 

Most past studies on the impact of paclitaxel exposure on neuroinflammation in the mouse or rat DRG have been in males. Like our findings in males shown here, others have shown increases in males following paclitaxel administration in *Aif1* in mice [29], *Gfap* in rats [30], *Ccl2* in rats [11], *Tlr4* in rats [7,10], and *Il6* in rats [31]. Our present work extends these findings to demonstrate that *Cxcl9* expression is also increased in the DRG in male mice following paclitaxel exposure, and we are unaware that this has been demonstrated previously. Specifically, this increase on Day 8 occurred at the same dose of paclitaxel that significantly altered mechanical sensitivity, and this equipotency did not occur between the behavioral measurement and most markers measured, suggesting that *Cxcl9* may be a more sensitive marker of the expression of pain in CIPN than the others. 

In female mice, we also found that *Aif1, Gfap, Ccl2, Tlr4, Il6*, and *Calca* expression were significantly increased in the DRG. Remarkably, paclitaxel exposure did not increase *Cxcl9* expression in female mouse DRG. Therefore, regarding the DRG, the main sex difference observed was in *Cxcl9*, where paclitaxel is very potent at increasing expression in the DRG of male but not female mice. Relative to other members of the cytokine family, the role of CXCL9 in pain models such as CIPN is underexplored. CXCL9 is one of several cytokines that can activate the CXCR3 receptor within the DRG and has recently been implicated in the modulation of neuropathic pain in male rodents [32] and cancer-related pain in female rodents [33]. 

The role of the PAG in pain regulation cannot be overstated. Ascending nociceptive afferents run through the PAG, and when descending, reciprocal regulatory control of the PAG modulates the dorsal horns of the spinal cord. However, compared with the DRG or spinal cord, a paucity of research has investigated whether CIPN is associated with functional changes in the PAG. In one study, a paclitaxel dosing regimen that increased thermal sensitivity also increased neuronal activity in the PAG in rats, and both phenomena were attenuated by treatment with gabapentin [34]. Another recent study evaluated the effect of paclitaxel on neuronal function in rats using MRI, and results suggested that paclitaxel-injected rats showed neuroplastic changes in the PAG that may be indicative of glial cell activity [35]. These results suggest that the PAG may be a novel target for treating CIPN. 

Ours is the first study we are aware of to demonstrate increases in neuroinflammation in the PAG associated with paclitaxel exposure. In contrast to what was observed in the DRG, we observed more changes in the PAG on Day 15 versus Day 8, and more changes in females than in males. In male mice, only the microglial marker was increased on Day 15. In female mice, the following genes were expressed in the PAG of paclitaxel-treated mice as compared with vehicle-treated mice: *Aif1* (Days 8 and 15), *Tlr4* (Day 15), *Ccl2* (Days 8 and 15), and *Calca* (Day 15). Microglial activation in the PAG has been associated with rodent models of other types of chronic pain [36]. Another study reported that PAG microglia are sexually dimorphic in their response to morphine or LPS-exposure, effects linked to the TLR4 receptor and increased sensitivity in female rodents [37]. CCL2 is another key activator of microglia, through the CCR2 receptor. CCL2 in the DRG has been shown to play a pivotal role in the peripheral mechanisms of neuropathic pain (see [38] for review). Interestingly, *Ccl2* expression was also significantly increased in the PAG following CCI or sciatic nerve injury [39]. Taken together, in our CIPN model, females were more sensitive to the effects of paclitaxel on the activation of microglia in the PAG which may involve signaling through CCR2 and/or TLR4 receptors. Lastly, intra-PAG CGRP has also been implicated in central pain sensitization, most notably in migraine models [40,41,42]. It is well established that migraines are more common in women, and emerging evidence suggests that sexually dimorphic CGRP signaling mechanisms may partially underlie this sex difference [43]. While it has been reported that CGRP is elevated in the DRG and dorsal spinal cord in rodent models of CIPN, this is the first report we are aware of making this observation in a female CIPN model. 

We find it important to note that sex-dependent increased potency of paclitaxel to produce mechanical sensitivity in male mice did not correspond in a similar pattern to neuroinflammatory effects in the DRG or PAG. Looked at in another way, in both males and females, mechanical sensitivity is often significantly increased at doses of paclitaxel that are not sufficient to produce measurable pro-inflammatory gene expression changes in the regions, times, and analytes tested in the present study. More research is warranted to elucidate whether one or both (behavior and neuroinflammation) outcome measures are more translationally relevant to CIPN in humans and, therefore, are better predictions of the beneficial effects of therapeutic interventions. 

The phytocannabinoid CBG significantly attenuated the development of mechanical sensitivity associated with paclitaxel administration in male and female mice. As mentioned above, we have previously used this prophylactic treatment strategy to determine that two other phytocannabinoids, Δ9-THC and CBD, can prevent the development of mechanical sensitivity associated with paclitaxel administration in male or female C57Bl/6 mice [23,24,25,26]. The varied and unique pharmacodynamic profile of CBG has also increased interest in its potential antinociceptive and/or anti-neuropathic effects. It was recently demonstrated that either acute or chronic injection of CBG can reverse an already established CIPN associated with cisplatin administration in male and female mice [27,28]. Receptor mechanisms associated with these antinociceptive effects include α-2 adrenergic and CB1 and CB2 receptor activation. The present results synthetize these findings to show that CBG also shows prophylactic efficacy in a CIPN model. In this experiment, we harvested the PAG and demonstrated that CBG administration also provided protection from increased expression of *Aif1*, *Ccl2*, *Calca*, and *Tlr4* in female mice, and we replicated our results, showing that paclitaxel exposure did not increase these inflammatory markers in the PAG of male mice. We are unaware of any previous reports assessing the effects of CBG treatment on these markers of pain and neuroinflammation. 

In summary, we demonstrate here the dose-, time course-, nervous system region-, and sex-specific effects of paclitaxel on mechanical sensitivity and neuroinflammation in C57Bl/6 mice, and an overall protective effect of prophylactic treatment with the non-psychoactive phytocannabinoid CBG. Sex-dependent effects on mechanical sensitivity, wherein males’ sensory thresholds are sensitized at lower concentrations of paclitaxel than females’, appear early but are not long lasting, and do not correspond in a similar pattern to neuroinflammatory effects. In the DRG, males and females are similarly sensitive early on to the neuro-inflammatory effects, but these effects dissipate earlier in males than in females. In the PAG, females show more neuro-inflammation and at later timepoints. Two analytes that demonstrate the most compelling sexual dimorphism are *Cxcl9* and *Ccl2* in males, and *Tlr4* in females. Future studies are warranted to explore how these sex differences may shed light on the mechanisms of CIPN and how non-psychoactive cannabinoids such as CBG may engage these targets to prevent or attenuate the effects of paclitaxel and other chemotherapeutic agents on the nervous system. 

## 4. Materials and Methods

### 4.1. Animals

The animal experiments presented in this article were approved by the Institutional Animal Care and Use Committee (IACUC) at Temple University (Philadelphia, PA, USA). A total of 240 mice were used in this study (*n* = 8/group). The mice used in these experiments were purchased from Taconic Biosciences (Germantown, NY, USA). They were C57BL/6J mice (male and female), aged 6–8 weeks at time of arrival to the vivarium. All mice acclimated to the vivarium for at least 5 days prior to initiation of behavioral testing. Mice were maintained in an enriched environment with a dark/light cycle of 12 h and a temperature of 22 °C. Mice were housed 4 per cage and had ad libitum access to regular food and water. The behavioral experiments on the mice were performed during the light cycle. Every effort was made to ensure optimal welfare conditions before, during, and after each experiment, and the mice were observed daily for their general condition. The size of the animal groups for the experiments was based on data from previous studies. The observer of the behavioral tests was not aware of the treatment of the animals. Mice were randomly assigned to their groups.

### 4.2. Drugs

Paclitaxel (Hospira, Lake Forest, IL, USA, NDC 61703-0342-22) was procured from Temple University Pharmacy. Paclitaxel was dissolved in a mixture of kolliphor (Sigma-Aldrich, St. Louis, MO, USA), ethanol (Sigma-Aldrich), and saline (mixture proportion 1:1:18). Intraperitoneal injections of paclitaxel were performed every other day, with four administrations in total, at a dose of 8–32 mg/kg for each injection. Control mice received the vehicle (1:1:18, ethanol, kolliphor, and saline) at a volume of 10 mL/kg, i.p. and followed the same injection schedule. CBG was purchased from Cayman Chemical, Ann Arbor MI. CBG was dissolved in a mixture of kolliphor (Sigma-Aldrich, St. Louis, MO, USA), ethanol (Sigma-Aldrich), and saline (mixture proportion 1:1:18) and was administered 30 min prior to each paclitaxel injection. Doses of paclitaxel and CBG were based on previous work from our and others’ laboratories [23,24,25,26,27,28].

### 4.3. Mechanical Sensitivity: Von Frey Filaments Test

Baseline mechanical sensitivity testing took place for three consecutive days (Days −2, −1, and 0) before administration of the first paclitaxel injection on Day 1. During baseline testing and then again of Day 7 or Day 14, mice were placed in individual Plexiglas compartments (Med Associates, St. Albans, VT, USA) on top of a wire grid floor suspended 20 cm above the laboratory bench top and acclimatized to the environment for 30 min before each test session. Mechanical allodynia was assessed using Von Frey monofilaments of varying forces (0.07–2.0 g) applied to the mid-plantar surface of the right hind paw, with each application held in c-shape for 6 s, starting with the 0.07 filament. If no response was elicited, the next filament was tested until a response was elicited. Filaments were then retested in descending order until the filament did not elicit a response, and the lowest filament to elicit a response was recorded [23,24,25,26].

### 4.4. Quantitative Real-Time Polymerase Chain Reaction (qRT-PCR)

DRG and PAG were dissected from mice on Day 8 or Day 15, and total RNA was extracted using a Quick-RNA™ MiniPrep kit (#R1054, Zymo Research, Irvine, CA, USA) according to manufacturer’s instructions. All RNA samples had A260/A280 ratios of 1.8 to 2.0. Purified RNA was treated with DNase I and eluted with DNase/RNase-free water. Reverse transcription was performed using an RT2 First Strand cDNA kit (#330401, Qiagen, Hilden, Germany) according to manufacturer’s instructions. Quantitative real-time PCR assays were performed using Taq Man Gene Expression Assays (Thermo Scientific, Waltham, MA, USA) to quantify mRNA levels of *Aif1* (for Iba-1), *Tlr4*, *Ccl2*, *Cxcl9*, *Il6*, *Gfap*, and *Calca* (CGRP), using the 18S ribosomal RNA probe as an internal control. The PCR assay was performed using a QuantStudio™ 3 Real-Time PCR System (Thermo Scientific), with the threshold cycle (Ct) calculated by QuantStudio 3 qPCR software. Each reaction was run in triplicate and contained 50–100 ng of RNA in a final reaction volume of 20 μL. Expression levels were calculated using the 2^−ΔΔCt^ method. Samples with a Ct value higher than 40 were deemed too low to be detected. 

### 4.5. Statistical Analyses

Data were checked for normality of distribution prior to statistical analyses using one- and two-way ANOVAs where indicated. Appropriate post hoc analyses stated in the text were performed to compare which treatment groups were statistically significantly different from the vehicle control groups. All statistical analyses were performed using GraphPad Prism 10.10. 

## Figures and Tables

**Figure 1 ijms-25-04277-f001:**
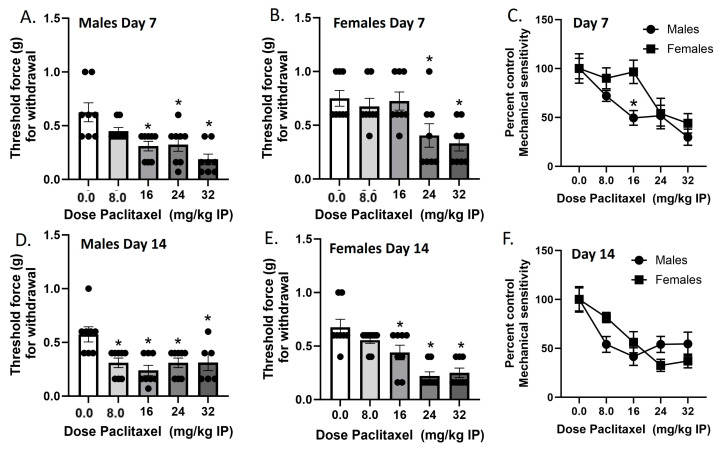
Paclitaxel lowers threshold for paw withdrawal in male (**A**,**D**) and female (**B**,**E**) C57Bl/6 mice. Mice are treated with vehicle or increasing doses of paclitaxel on Days 1, 3, 5, and 7 in a between-subjects design (*n* = 8/group). Mechanical sensitivity is measured on Days 7 (**A**–**C**) and 14 (**D**–**F**). Results demonstrate that a higher dose of paclitaxel is required to significantly lower threshold to response to tactile stimulation using Von Frey filaments on Day 7 (**C**) but not Day 14 (**F**). (**A**,**B**,**D**,**E**) One-way ANOVAs with Dunnett’s multiple comparison tests, * = *p* < 0.05 as compared to vehicle control. (**C**,**F**) Two-way ANOVAs with Sidak’s multiple comparison test, * = *p* < 0.05.

**Figure 2 ijms-25-04277-f002:**
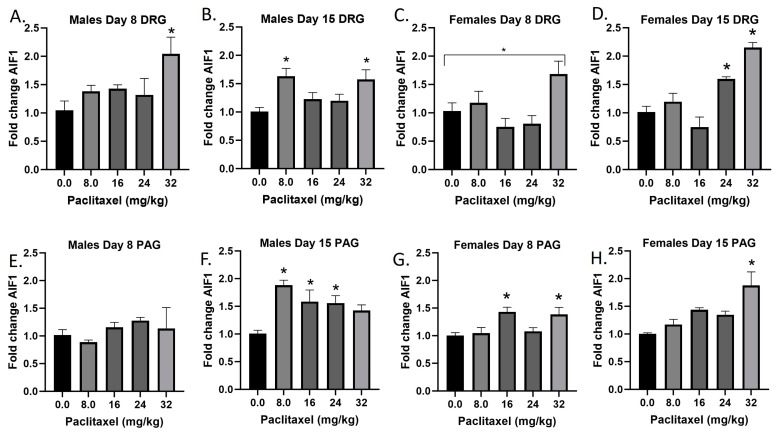
Paclitaxel increases expression of Aif1 in DRG (**A**–**D**) and PAG (**E**–**H**) of male and female C57Bl/6 mice. Mice are treated with vehicle or increasing doses of paclitaxel on Days 1, 3, 5, and 7 in a between-subjects design (*n* = 3–5/group). Mice are euthanized on Day 8 or 15 and tissues are harvested for RT-PCR analysis. One-way ANOVAs with Dunnett’s multiple comparison tests; * = *p* < 0.05 as compared with saline.

**Figure 3 ijms-25-04277-f003:**
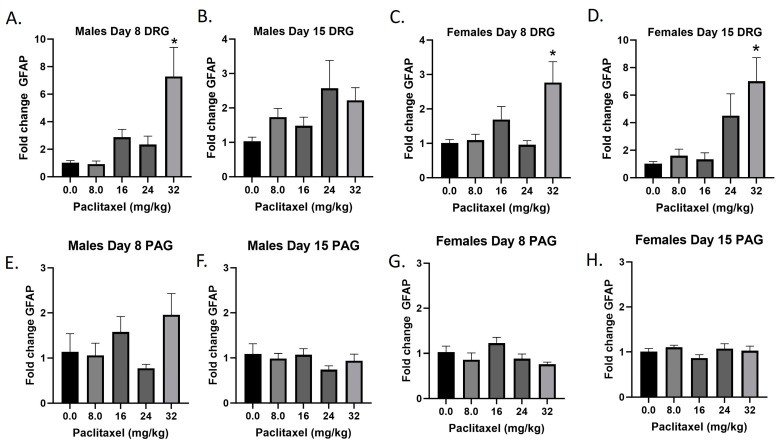
Paclitaxel increases expression of *Gfap* in DRG (**A**–**D**) but not PAG (**E**–**H**) in male and female C57Bl/6 mice. Mice are treated with vehicle or increasing doses of paclitaxel on Days 1, 3, 5, and 7 in a between-subjects design (*n* = 3–5/group). Mice are euthanized on Day 8 or 15 and tissues are harvested for RT-PCR analysis. One-way ANOVAs with Dunnett’s multiple comparison tests; * = *p* < 0.05 as compared with saline.

**Figure 4 ijms-25-04277-f004:**
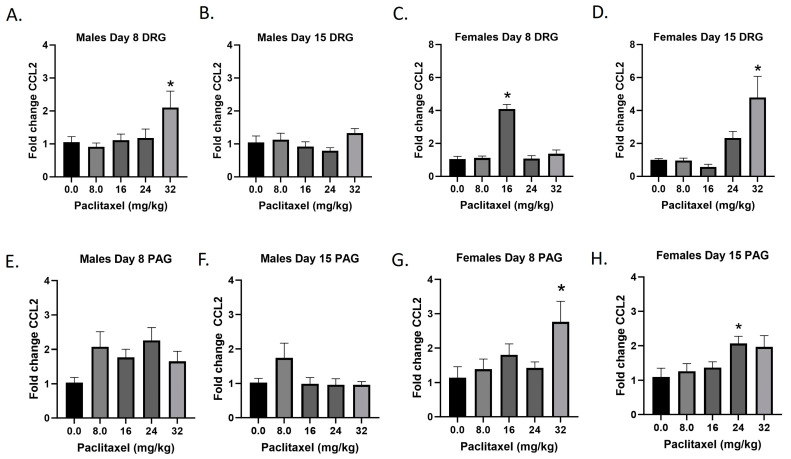
Paclitaxel increases expression of *Ccl2* in DRG (**A**–**D**) and PAG (**E**–**H**) in male and female C57Bl/6 mice. Mice are treated with vehicle or increasing doses of paclitaxel on Days 1, 3, 5, and 7 in a between-subjects design (*n* = 3–5/group). Mice are euthanized on Day 8 or 15 and tissues are harvested for RT-PCR analysis. One-way ANOVAs with Dunnett’s multiple comparison tests; * = *p* < 0.05 as compared to vehicle control.

**Figure 5 ijms-25-04277-f005:**
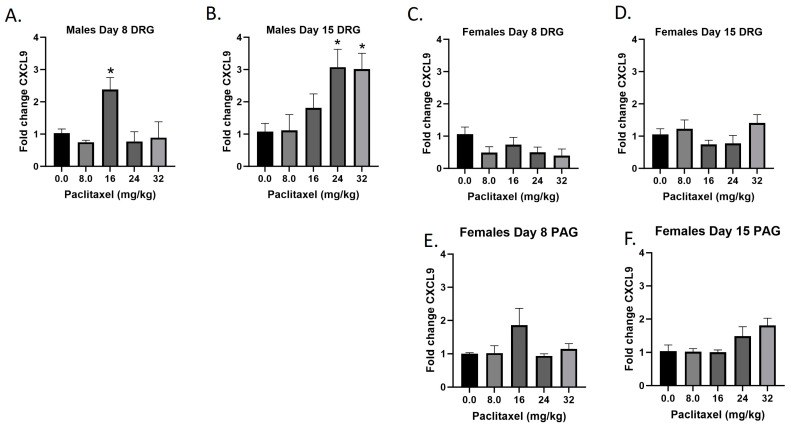
Paclitaxel increases expression of *Cxcl9* in DRG (**A**–**D**) but not PAG (**E**,**F**) in male but not female C57Bl/6 mice. Mice are treated with vehicle or increasing doses of paclitaxel on Days 1, 3, 5, and 7 in a between-subjects design (*n* = 3–5/group). Mice are euthanized on Day 8 or 15 and tissues are harvested for RT-PCR analysis. *Cxcl9* was not detected in the PAG of male mice. One-way ANOVAs with Dunnett’s multiple comparison tests; * = *p* < 0.05 as compared to vehicle control.

**Figure 6 ijms-25-04277-f006:**
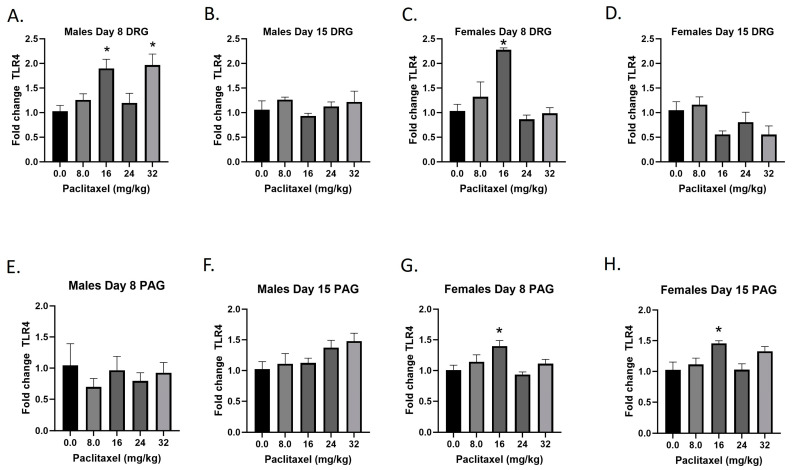
Paclitaxel increases expression of *Tlr4 i*n DRG (**A**–**D**) and PAG (**E**–**H**) in male and female C57Bl/6 mice. Mice are treated with vehicle or increasing doses of paclitaxel on Days 1, 3, 5, and 7 in a between-subjects design (*n* = 3–5/group). Mice are euthanized on Day 8 or 15 and tissues are harvested for RT-PCR analysis. One-way ANOVAs with Dunnett’s multiple comparison tests; * = *p* < 0.05 as compared to vehicle control.

**Figure 7 ijms-25-04277-f007:**
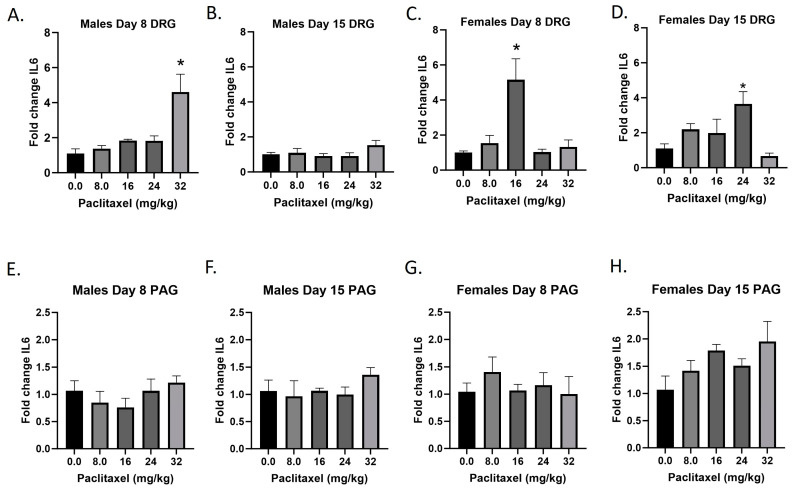
Paclitaxel increases expression of IL-6 in DRG (**A**–**D**) and PAG (**E**–**H**) male and female C57Bl/6 mice. Mice are treated with vehicle or increasing doses of paclitaxel on Days 1, 3, 5, and 7 in a between-subjects design (*n* = 3–5/group). Mice are euthanized on Day 8 or 15 and tissues are harvested for RT-PCR analysis. One-way ANOVAs with Dunnett’s multiple comparison tests; * = *p* < 0.05 as compared to vehicle control.

**Figure 8 ijms-25-04277-f008:**
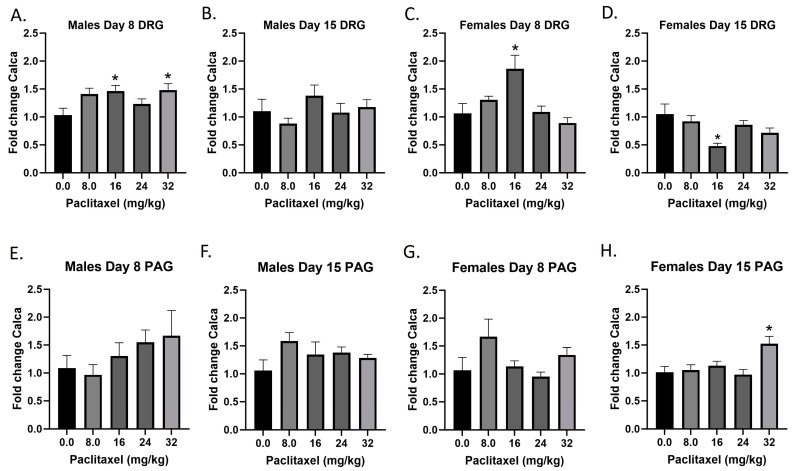
Paclitaxel increases expression of CGRP in the DRG (**A**–**D**) and PAG (**E**–**H**) in male and female C57Bl/6 mice. Mice are treated with vehicle or increasing doses of paclitaxel on Days 1, 3, 5, and 7 in a between-subjects design (*n* = 3–5/group). Mice are euthanized on Day 8 or 15 and are harvested for RT-PCR analysis. One-way ANOVAs with Dunnett’s multiple comparison tests; * = *p* < 0.05 as compared to vehicle control.

**Figure 9 ijms-25-04277-f009:**
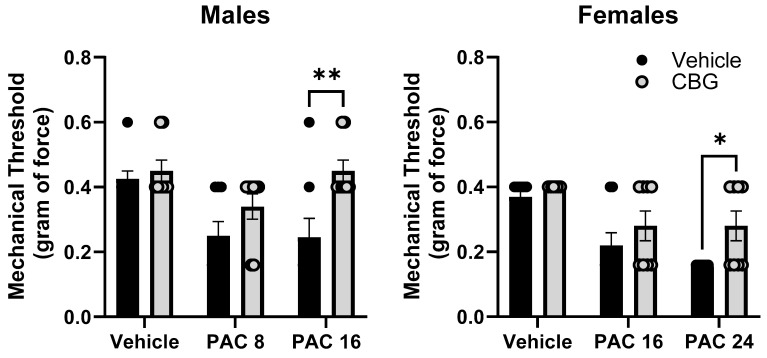
Prophylactic CBG administration prevents mechanical sensitivity associated with paclitaxel exposure in male (**left**) and female (**right**) C57Bl/6 mice. Mice are treated with vehicle or CBG (10 mg/kg i.p.) 30 min prior to vehicle or paclitaxel (8.0 or 16 mg/kg in males, 16 or 24 mg/kg i.p. in females,) on Days 1, 3, 5, and 7 in a between-subjects design (*n* = 8/group). Mechanical sensitivity is measured on Days 0 and 14. Results demonstrate that prophylactic administration of CBG protected the male and female mice from increased mechanical sensitivity. Two-way ANOVAs with Sidak’s multiple comparison tests; * = *p* < 0.05 as compared to vehicle control. ** = *p* < 0.01.

**Figure 10 ijms-25-04277-f010:**
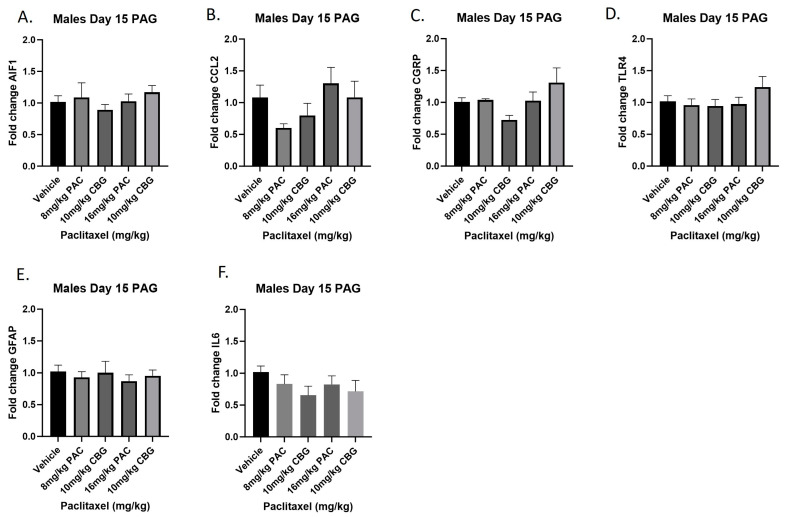
Prophylactic CBG administration prevents increased expression of inflammation and pain markers (**A**–**F**) in the PAG of male C57Bl/6 mice. Mice are treated with vehicle or CBG (10 mg/kg i.p.) 30 min prior to vehicle or paclitaxel (8.0 or 16 mg/kg i.p.) on Days 1, 3, 5, and 7 in a between-subjects design (*n* = 3–5/group). Mice are euthanized on Day 15 and PAGs are harvested for RT-PCR analysis. Results demonstrate that paclitaxel exposure does not alter gene expression in the PAG.

**Figure 11 ijms-25-04277-f011:**
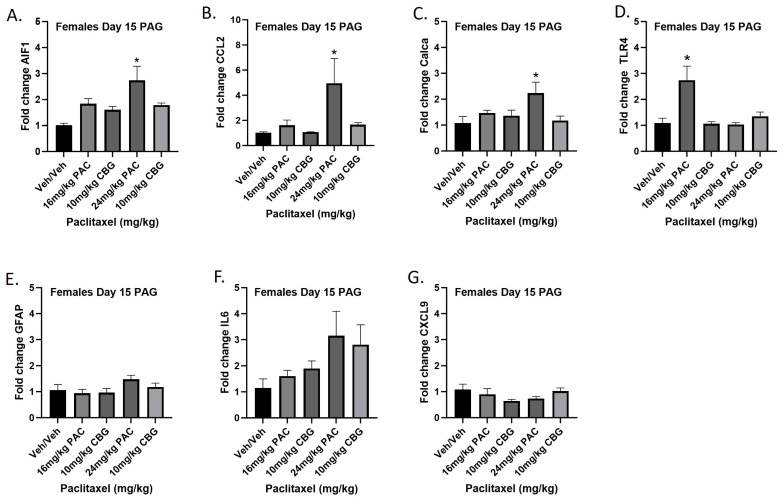
Prophylactic CBG administration prevents increased expression of inflammation and pain markers (**A**–**G**) in the PAG of female C57Bl/6 mice. Mice are treated with vehicle or CBG (10 mg/kg i.p.) 30 min prior to vehicle or paclitaxel (16 or 24 mg/kg i.p.) on Days 1, 3, 5, and 7 in a between-subjects design (*n* = 3–5/group). Mice are euthanized on Day 15, and PAGs are harvested for RT-PCR analysis. Results demonstrate that CBG administration prevented increased expression of *Aif1*, *Ccl2*, *Calca*, and *Tlr4* associated with paclitaxel exposure. One-way ANOVAs with Dunnett’s multiple comparison tests; * = *p* < 0.05.

**Table 1 ijms-25-04277-t001:** Summary of neuroinflammatory changes in the DRG and PAG of male and female mice at Day 8 or Day 15 following paclitaxel exposure. Grey/green hatched shading represent significant changes in gene expression that occurred at the same doses that produced significant changes in mechanical sensitivity. Green shading indicates significant increase in gene expression compared with vehicle control, but at a higher dose than that required to produce mechanical sensitivity. No shading represents no significant change in gene expression at any dose tested. Minus symbol represents significant decrease in gene expression. ND = gene not detected.

	DRG				PAG			
	Males		Females		Males		Females	
	D8	D15	D8	D15	D8	D15	D8	D15
** *AIF1* **								
** *GFAP* **								
** *CCL2* **								
** *CXCL9* **					**ND**	**ND**		
** *TLR4* **								
** *IL6* **								
** *Calca* **				**-**				

**Table 2 ijms-25-04277-t002:** Statistical summary of sex differences in neuroinflammatory changes in the DRG and PAG of male and female mice at Day 8 or Day 15 following paclitaxel exposure. Two-way ANOVAs with Sex and Dose Paclitaxel as factors. Shown are F and *p* values for Sex as a factor. NS = not statistically significant. ND = not detected.

Day 8 DRG			Day 8 PAG		
AIF1	F_(1, 40)_ = 11.17	*p* = 0.0018	AIF1	F_(1,40)_ = 1.619	NS
GFAP	F_(1,40)_ = 10.26	*p* = 0.0021	GFAP	F_(1,40)_ = 6.697	*p* = 0.0132
CCL2	F_(1,40)_ = 10.91	*p* = 0.0020	CCL2	F_(1,40)_ = 0.06438	NS
CXCL9	F_(1,40)_ = 14.47	*p* = 0.0005	CXCL9	ND in males	
TLR4	F_(1,40)_ = 2.895	NS	TLR4	F_(1,40)_ = 5.848	*p* = 0.0200
IL6	F_(1,40)_ = 0.2003	NS	IL6	F_(1,40)_ = 1.360	NS
Calca	F_(1,40)_ = 0.9614	NS	Calca	F_(1,40)_ = 0.3793	NS
**Day 15 DRG**			**Day 15 PAG**		
AIF1	F_(1,40)_ = 0.03062	NS	AIF2	F_(1,40)_ = 2.425	NS
GFAP	F_(1,40)_ = 8.510	*p* = 0.0058	GFAP	F_(1,40)_ = 0.4725	NS
CCL2	F_(1,40)_ = 10.21	*p* = 0.0027	CCL2	F_(1,40)_ = 7.670	*p* = 0.0085
CXCL9	F_(1,40)_ = 23.76	*p* < 0.0001	CXCL9	ND in males	
TLR4	F_(1,40)_ = 10.66	*p* = 0.0023	TLR4	F_(1,40)_ = 0.2123	NS
IL6	F_(1,40)_ = 13.82	*p* = 0.0006	IL6	F_(1,40)_ = 16.08	*p* = 0.0004
Calca	F_(1,40)_ = 13.32	*p* = 0.0008	Calca	F_(1,40)_ = 5.335	*p* = 0.0261

## Data Availability

The data presented in this study are available on request from the corresponding author.

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
