# Peer review of "Paclitaxel-Associated Mechanical Sensitivity and Neuroinflammation Are Sex-, Time-, and Site-Specific and Prevented through Cannabigerol Administration in C57Bl/6 Mice"

_ijms, 2024, doi:10.3390/ijms25084277_

Round 1

Reviewer 1 Report

Comments and Suggestions for Authors

The present study wants to evaluate possible mechanisms underlying the development of chemoterapy-induced pain.

Data can be interesting and the work done is important, also considering the effort to carry out all experiments in male and female subjects.

However the manuscript is not well organized:

abstract... need to be divided in sessions.. it looks like an introduction

introduction: all data about the different aspects of the research need to be included here, not in the results... no methods in this session!

results need to be clearly written with more tables if possible

Discussion need to be shortened avoiding the repetition of the results

Comments on the Quality of English Language

There are some spelling mistakes

Author Response

  1. We have divided the abstract and improved it
  2. We moved relevant items to intro versus methods versus results
  3. we removed some of the repeated results from the discussion

Reviewer 2 Report

Comments and Suggestions for Authors

Paper titled (Sex- and site- specific paclitaxel-associated mechanical sensitivity and neuroinflammation prevention by cannabigerol in male and female C57Bl/6 mice) explored the effect of cannabigerol in alleviating paclitaxel induced mechanical sensitivity and neuroinflammation  and determined sex & site specificity for this protective effect. 

I find this work is overestimated and study design is not appropriate including many many overlapping factors do not enable a scientist to take a correct conclusion, here are the specific reasons also.

1- title: is giving information not directly! please try to formulate it better & to be more informative about what did authors find? 
Also you mentioned (sex specificity) in the title & hence no need to mention (male and female)

2- Abstract: should be amended by some numerical data from the results

3- Define each abbreviation at the first appearance in the abstract and also in the body text. For example (PAG) was not defined in the abstract

4- revise if the number of key words fits the journal guidelines

5- What was the rational for this study? what was the novelty or research gap? the idea is complex relating many factors (sensitivty, response to treatment in relation to sex & site) many overlapping factors. And also time!!!

Oh, also different doses of paclitaxel are used!!!!

6- The aim of the work at teh end of intorduction resembles a methodology. Authors should present the aim & how they achieved it

7-  In results, male and female animals were compared separately & no statistical analysis compared them together, hence we cannot make a cocnlusion which sex was more susceptible to paclitaxel effect or cannabingol

Also comparsion at each time was given sepaartely, so what isi the value of the time course experiment?

8- Use appropriate abbreviations for minutes, seconds...etc

9- Mention "n" in each illustation individually
10-Every abbreviation in figures should be explained in the figure legend to be self explanatory & stands alone.

11- Animal housing conditions should be described clearly, cages, number of animal per cage, conditions, how minimized animal suffering?
12- Authors should give the source of chemicals, kits and antibodies completely and consistently (code, company, town, state and country) & version for software

13- The data in Table 1 should be given in column charts as supplementary materials

14-  Table 1 should come after figures indicate the exact results 

15- In figure legends (*=p<0.05) means compared to what group? why this group only was compared to others? all groups should be compared

16- Using one way ANOVA in data in Figure 9 is NOT appropriate as more than 1 factor is influencing the results

17- Give the detailed forces for the filaments

18- Authors tested allodynia once? or took an average of many trials?

19- Mention the details of kits mentioned in methods

20- Authors have to check the normality of distribution of the results by a suitable post hoc test (such as Shapiro-Wilk test or K-S test) before deciding to choose certain ANOVA. If the normality test indicated normal dist of the data, so use one-way ANOVA, if not, use non parametric ANOVA. In all cases choose a suitable post-hoc test
21- Authors should confirm in methods that "every possible comparison between the study groups was considered" and apply this in results.

22- Discussion is not usually numbered like that

23- Mention references for the doses and schedules.
24- What was the refernce for PCR calculation by the Ct method?

25- Methods in general lacks refernces at many occasions

Comments on the Quality of English Language

fine

Author Response

1) We edited the title. Because we observed specific effects on sex, time, and brain region, it was impossible to state all findings in the title

2) We significantly edited the abstract

3) We defined abbreviations

4) I think this was in the guidelines of the journal

5) We have clearly stated the rationale which addresses the research gap that sex, time course, and site specific changes in neuroinflammation associated with CIPN have not been adequately addressed and that the best way to do these comparisons is all side by side in one comprehensive study.

6) We have fixed the introduction

7) For behavior, males and females have been compared together with two-way Anova in Figure 1. Because we observe sex differences in the potency of paclitaxel in males and females in the first experiment, we did not feel it was appropriate in the rest of the experiments to compare by sex when we knew that the potency of paclitaxel was different in males versus females.

8) We fixed this

9) This is addressed

10) We believe this is the case

11) This was stated in the methods

12) This was stated in the methods

13) We have not changed the Table. The other reviewer wanted more charts in the text, so because of the opposite recommendations by the two reviewers, we have left the Table as is except moved its location.

14) We did this

15) We added this

16) We changed this based on your suggestion

17) This was given. These are standard von Frey filaments used widely in the field.

18) This is stated in the methods

19) This is included

20) This is included in the statistics section

21) We do not believe this is appropriate

22) This has been changed

23) Doses and schedules are consistent with several CIPN studies that we cite throughout the manuscript, including our own prior work

24) This is referenced

25) These are standard methods we have published on and our references are included throughout the manuscript.  

Round 2

Reviewer 1 Report

Comments and Suggestions for Authors

The manuscript was improved but it still need to be changed, in particular:

- in the abstract results cannot be included with this detail, please summarize it 

- in the figure use always the same format: in some the sex of the animal is written below in other it is not clear

- check carefully the figure to mach result/name of the graphic

Author Response

We have edited the abstract, corrected Figure 11, and made sure all of the legends were correct. Thank you!

Reviewer 2 Report

Comments and Suggestions for Authors

The revised version of paper titled (Sex- and site- specific paclitaxel-associated mechanical sensitivity and neuroinflammation prevention by cannabigerol in male and female C57Bl/6 mice) was very partly improved in some parts in the methods. However results are still confusing and Statistical analysis was not appropriate or justified (see my original revision)

Also n value was not provided in all figure legends,  

The main issue herein is that statistical analysis was done separately and no evidence that it was sex specific or otherwise. To give such a conclusion, all groups (both dex) and so on. should be compared within the same ANOVA & posy hoc analysis. 

Sorry the conclusion given here is NOT correct & data do not support the conclusion

Also what was the refernce for drug doses & for the VOn Frey filament tests

Comments on the Quality of English Language

Mild

Author Response

We have now conducted two-way anova for each gene at each timepoint and site. We have reported the F and p values in a table for the significant effects of sex at each timepoint and site. These results support our conclusions.

Also n value was not provided in all figure legends, 

-- We do not see where they are missing. 

Also what was the refernce for drug doses & for the VOn Frey filament tests

--we added the references

Round 3

Reviewer 2 Report

Comments and Suggestions for Authors

The statistics were adjusted to some extent, please be acreful in the future 

Comments on the Quality of English Language

thanks

Author Response

We really appreciate the suggestion and believe the changes really strengthened the conclusion, thank you!